# Comparison of Three Skin Sampling Methods and Two Media for Culturing *Malassezia* Yeast

**DOI:** 10.3390/jof6040350

**Published:** 2020-12-09

**Authors:** Abdourahim Abdillah, Saber Khelaifia, Didier Raoult, Fadi Bittar, Stéphane Ranque

**Affiliations:** 1Institut de Recherche pour le Développement, Aix Marseille Université, Assistance Publique-Hôpitaux de Marseille, Service de Santé des Armées, VITROME: Vecteurs—Infections Tropicales et Méditerranéennes, 19-21 Boulevard Jean Moulin, 13005 Marseille, France; abdourahim15@live.fr; 2IHU Méditerranée Infection, 19-21 Bd Jean Moulin, 13005 Marseille, France; khelaifia_saber@yahoo.fr (S.K.); didier.raoult@mediterranee-infection.com (D.R.); fadi.BITTAR@univ-amu.fr (F.B.); 3Institut de Recherche pour le Développement, Aix Marseille Université, Assistance Publique-Hôpitaux de Marseille, Service de Santé des Armées, MEPHI: Microbes, Evolution, Phylogénie et Infection, 19-21 Boulevard Jean Moulin, 13005 Marseille, France

**Keywords:** *Malassezia*, isolation, culture media, skin sampling

## Abstract

*Malassezia* is a lipid-dependent commensal yeast of the human skin. The different culture media and skin sampling methods used to grow these fastidious yeasts are a source of heterogeneity in culture-based epidemiological study results. This study aimed to compare the performances of three methods of skin sampling, and two culture media for the detection of *Malassezia* yeasts by culture from the human skin. Three skin sampling methods, namely sterile gauze, dry swab, and Transwab^TM^ with transport medium, were applied on 10 healthy volunteers at 5 distinct body sites. Each sample was further inoculated onto either the novel FastFung medium or the reference Dixon agar for the detection of *Malassezia* spp. by culture. At least one colony of *Malassezia* spp. grew on 93/300 (31%) of the cultures, corresponding to 150 samplings. The positive culture rate was 67%, 18%, and 15% (*P* < 10^−3^), for samples collected with sterile gauze, Transwab^TM^, and dry swab, respectively. The positive culture rate was 62% and 38% (*P* < 0.003) by using the FastFung and the Dixon media, respectively. Our results showed that sterile gauze rubbing skin sampling followed by inoculation on FastFung medium should be implemented in the routine clinical laboratory procedure for *Malassezia* spp. cultivation.

## 1. Introduction

The *Malassezia* yeasts are basidiomycetes of the Ustilaginomycotina subphylum and Malasseziomyctes class [1]. These yeasts are lipid-dependent because comparative genomic analyses have revealed that all *Malassezia* lack the gene encoding the synthesis of fatty acids [2]. They are part of the normal eukaryotic microbiota of the skin in humans and other warm-blooded animals, and it is estimated that about 75–98% of healthy subjects carry these yeasts [3]. However, under certain circumstances, *Malassezia* species can be involved in relatively common skin diseases, including pityriasis Versicolor, seborrheic dermatitis, and folliculitis. They also cause sporadic severe systemic infections in neonates and immunocompromised individuals with parenteral nutrition [4,5,6].

*Malassezia* spp. are not able to grow on the mycological media routinely used in the clinical laboratory because of their lipid-dependency; they necessitate using specific culture media, such as Dixon agar or modified Dixon agar, Leeming–Notmann agar or modified Leeming–Notmann agar, etc. Yet specific media are relatively seldom used and partially explain why *Malassezia* spp. has been detected via metagenomics but not via culture, especially in the human gut [7,8] or respiratory tract [9,10]. Several formulations of the media are used in many laboratories, such as CHROMagar *Malassezia*^TM^ or Tween 60-esculin agar but there is no consensus on the culture medium to be used to isolate these yeast species. Moreover, the lack of standardization of skin sampling procedure and culture media used to grow these fastidious yeasts is a source of heterogeneity in culture-based study results [5,11], that should be avoided. Today the use of several methods of skin sampling are reported in the literature, including scraping, swabbing, and tape stripping. These heterogeneous sampling methods generate culture results heterogeneity and may cause a wide range of variation in the assessment of *Malassezia* yeast frequency and abundance. In this respect, many authors consider that the discrepant and heterogeneous findings of culture-based epidemiological studies are mainly due to the heterogeneity of the sampling procedure that has been used [12]. Probably, using heterogeneous *Malassezia* culture media is also a source of heterogeneity of culture based-studies’ findings. To address these limitations, this study aimed to compare the performances of three methods of skin sampling, namely sterile gauze, dry swab, and Transwab^TM^ (swab with transport medium), and two culture media, the FastFung medium, and the reference Dixon agar, for the detection of *Malassezia* yeasts by culture from the human skin.

## 2. Materials and Methods

### 2.1. Study Participants

A total of 10 healthy volunteers (9 males and 1 female, aged from 25–56 years; mean age: 31.7 years; median age: 27.5 years) at the IHU-MI (Institut Hospitalo-Universitaire Méditerranée Infection, Marseille, France) were included. The authors confirm that the ethical policies of the journal, as noted on the journal’s author guidelines page, have been adhered to and the appropriate ethical review committee approval has been received. All participants gave their informed consent and completed a form with age, gender whether or not systemic or local antifungal treatment has been taken in the last 15 days after sampling. Participants with a history of antifungal treatment were not included. This study protocol was approved by the CPP Ile de France II (N°19.05.29.69947 RIPH3) the 21 October 2019. Each volunteer included gave informed written consent to participate in the study.

### 2.2. Culture Media

The Dixon agar (Mediaproducts BV, Groningen, The Netherlands) containing 0.2 g/L chloramphenicol was used, as the reference medium for *Malassezia* spp. detection by culture. The FastFung medium (4.3% Schædler agar, 2% peptone, 1% glucose, 1% malt extract, 0.5% ox-bile, 0.5% Tween 60, 0.2% oleic acid and 0.25% glycerol [pH6], each from Sigma-Aldrich, Saint-Quentin Fallavier, France), developed at the Institut Hospitalo-Universitaire Méditeranée Infection (IHU-MI), has been derived from the Schædler agar as previously described [13] (Bittar et al. submitted).

### 2.3. Sample Collection and Cultures

Skin samples were collected at five different body sites (left and right nasolabial folds, left and right retroauricular folds, and presternal skin) using three sampling methods: dry cotton swabs (DCS) (Greiner Bio-One, Courtaboeuf, France), swabs with Amies liquid transport medium (Transwab^®^, ELITech, Puteaux, France); sterile gauze (non-woven sterile swabs 10 cm × 10 cm, Laboratoire SYLAMED, Paris, France). The three sampling methods were applied successively in a random order, which was noted, at each sampling site for 5 to 10 s. A total of 15 samples were collected for each participant. Each sample was then plated in parallel, in a random order that was noted, on Dixon agar (Mediaproducts BV) and FastFung medium supplemented with 0.1 g/L vancomycin (Sandoz, Levallois-Perret, France) and 0.03 g/L colistin (Sigma-Aldrich). The gauze was directly applied onto the plate. Unused dry cotton swabs, Transwab^®^, and sterile gauze were also plated on both media as negative controls. All plates were incubated aerobically at 30 °C for one week and examined daily for *Malassezia* spp. growth.

### 2.4. Colony Identification

Colonies of *Malassezia* spp. were identified via MALDI-TOF Mass Spectrometry (MALDI Biotyper, Bruker Daltonics, Bremen, Germany) with a reference spectra library supplemented with in house (including *Malassezia* spp.) reference spectra, following a previously published procedure [14]. The identification procedure was carried out on 1 to 3 colonies per plate.

### 2.5. Statistical Analysis

Statistical analyses were done with the SAS 9.4 software (SAS Institute Inc., Cary, NC, USA). The effect of different sampling and culture conditions on the number of CFUs (Colony Forming Units) of *Malassezia* spp. was tested with a negative binomial regression generalized linear model (Proc. Genmod), using generalized estimating equations to account for the non-independence of the measurements carried out on the same participant. The influence of the variables on the positivity (presence/absence) of the *Malassezia* spp. culture was tested by logistic regression (Proc. Logistics) allowing on the participant effect. Two-sided tests were used; and a *P-value* < 0.05 was considered statistically significant.

## 3. Results

A total of 93/300 (31%) of cultures, corresponding to 150 samplings from 5 distinct body sites, were positive for *Malassezia*, i.e., allowed the growth of at least one colony of *Malassezia* spp. We found a statistically significant independent effect (*P* < 10^−3^) of the sampling methods (Figure 1A); with the highest performance for sterile gauze totaling 67% (62/93) positive cultures compared to 18% (17/93) with Transwab^TM^ and 15% (14/93) with the dry swab. We also found a statistically significant (*P* < 0.003) independent effect of the culture medium, with the highest performance of the FastFung medium with 62% (58/93) of positive cultures compared to 38% (35/93) on Dixon agar medium (Figure 1B).

A total of 1082 colonies of *Malassezia* spp. were isolated and identified by MALDI-TOF Mass Spectrometry, including 455 (42%) *M. globosa*, 424 (40%) *M. sympodialis*, and 203 (18%) *M. restricta*. In some cases, we observed co-cultures between species such as *M. restricta* and *M. globosa*, *M. globosa* and *M. sympodialis* or *M. restricta* and *M. sympodialis*. Species isolation time ranged from 2 to 7 days for both media with an average of 4 days and no difference was observed between the two media. Noteworthy, the majority of the colonies of *M. globosa* and *M. restricta*, which are species of clinical interest and fastidious culture, were isolated on the FastFung medium (Figure 1C). The distribution of positive cultures according to sampling sites was 27/93 (29%), 49/93 (52.7), and 17/93 (18.3%) for the chest, ears (left and right retroauricular folds), and nose (left and right nasolabial folds), respectively. The isolated species were heterogeneously distributed according to sampling body sites, as illustrated in Figure 1D. The colony-forming unit frequency according to the sampling sites were 527, 490, and 65, for the chest, ear, and nose, respectively.

## 4. Discussion

Our findings demonstrate that rubbing the skin with a sterile gauze that is further inoculated onto a FastFung plate is the most efficient method for cultivating *Malassezia* yeast from the skin. Whereas a growing number of fungal metagenomics studies highlight a strong representation of *Malassezia* spp. among the fungal communities of various microbiomes, comparative studies of skin sampling techniques, and culture-based *Malassezia* spp. isolation media are lacking. Whereas the standardization of an effective and simple *Malassezia* spp. isolation procedure is crucial to enhance the patients’ diagnosis in the clinical laboratory and to homogenize and allow the comparison of distinct epidemiological studies results, to our knowledge, no previous study has been designed to optimize both sampling and culture procedures for *Malassezia* spp. isolation.

In the present study, we compared three skin sampling techniques (2 by swabbing and 1 by rubbing with a sterile gauze) on healthy volunteers and we found that rubbing with a sterile gauze was the most efficient method for recovering *Malassezia* yeast from the skin for culture, with a four times greater (67%) positive culture rate by using sterile gauzes than that by using either dry swabs (15%) or Transwab^TM^ (18%) (Figure 1A). Swabbing has been used in several studies [3,15,16] and it is considered as an alternative option to other methods such as scraping. Our findings question the reliability and efficacy of the swabbing technique not only in epidemiological studies but also for the diagnosis of patients in whom a *Malassezia* yeast infection is suspected. Noteworthy, our results regarding the effectiveness of sampling via sterile gauze are in line with those of Ilahi et al. [17], who used a similar sampling technique and found an overall positive culture rate of 75.71% on 70 patients, including pityriasis versicolor patients (28/28); breeders, most of them presenting skin lesions (15/17), and healthy controls (10/25). The sterile gauze technique presents many advantages, including simplicity, practicability, efficacy, cost-effectiveness, and non-invasiveness.

Furthermore, the FastFung medium proved more efficient than the reference Dixon medium for isolation and growth of *Malassezia* yeast, with 62% vs. 38% of positive culture rates, respectively (Figure 1B). Our 38% positive culture rate with the Dixon medium was similar to the one observed by Ilahi et al. [17], who found a 40% positive culture in healthy controls with the modified Dixon agar. This indicates that the difference observed in the present study is chiefly explained by the relatively enhanced efficacy of the FastFung medium. Noteworthy, the FastFung medium efficiently grew *M. globosa* and *M. restricta* (Figure 1C), both species that are considered to be among the most fastidious yeast to culture within the *Malassezia* genus [17,18,19,20]. Moreover, several studies showed that *M. globosa* and *M. restricta* are the most frequently involved species in pityriasis versicolor and seborrheic dermatitis, respectively [16,21,22]. The enhanced performances of the FastFung medium compared to the Dixon medium may at least in part be due to some components of the Schaedler agar. In line with this hypothesis was that our previous finding that this medium showed an enhanced efficiency to grow and isolate a variety of clinical fungi compared to the Sabouraud reference medium (Bittar et al., submitted).

*Malassezia* is implicated in most common skin infections including pityriasis versicolor and is associated with some chronic inflammatory diseases, including psoriasis and atopic dermatitis. Regarding psoriasis, for example, *Malassezia* has been shown to invade keratinocytes and induce overproduction transforming growth factor-beta, integrin chains, and heat shock protein 70, all of which are molecules involved in cell migration and hyperproliferation [23]. The improvement of sampling and culture techniques is a subject of great interest in diagnostics for better management.

Whereas the fact that the population of this study consisted only of healthy individuals might be considered as a limitation, we rather deem that this is one strength of our study. Indeed, *Malassezia* yeasts are relatively more abundant in diseased than in healthy conditions [24,25], thus cultivation of these yeasts from pityriasis versicolor of other *Malassezia*-associated skin diseases is probably easier than cultivating less abundant commensal yeasts from healthy skin samples. Another limitation of our study was the relatively small number of individuals included in the study. However, despite this limitation achieved, our study design reached sufficient statistical power to highlight statistically significant differences in comparing both sampling and culture methods results.

## 5. Conclusions

The findings of this study show that sterile gauze is an effective and reliable sampling technique for recovering *Malassezia* yeasts from the human skin for culture and that the efficacy for *Malassezia* culture of the novel FastFung medium is enhanced when compared to the reference Dixon medium. We thus propose implementing sterile gauze rubbing skin sampling followed by inoculation on FastFung medium in the routine clinical laboratory procedure for *Malassezia* spp. cultivation in both clinical and epidemiological studies.

## Figures and Tables

**Figure 1 jof-06-00350-f001:**
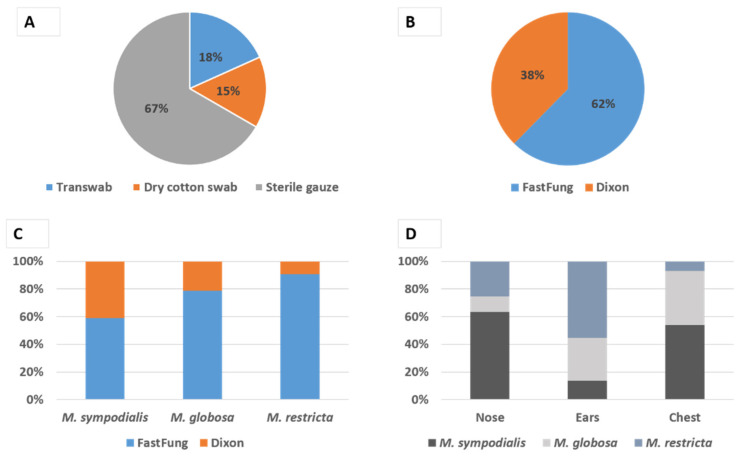
*Malassezia* species isolation. Relative contribution to *Malassezia* spp. positive culture of each sampling technique (**A**) and isolation media (**B**). (**C**) Relative contribution (CFU) of the Dixon agar and FastFung medium to the isolation of each detected *Malassezia* species. (**D**) Relative abundance (CFU) of each detected *Malassezia* species in the positive culture according to the body site sampled.

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
