# Peer review of "Comparison of Three Skin Sampling Methods and Two Media for Culturing Malassezia Yeast"

_jof, 2020, doi:10.3390/jof6040350_

Round 1
Reviewer 1 Report
‘Comparison of three skin sampling methods and two media for culturing Malassezia yeast’ by Abdillah et al.
The study is well written and interesting and focus on optimal sampling technique and testing a new culture media:
Regarding optimal sampling technique:
The authors are testing three methods sterile gauze, dry swab and Transwab and finds that the sterile gauze has a significant higher culture rate than the two others.
- Did the author consider that the area of rubbing with a sterile gauze might be larger than the area swabbed with the ‘dry swab’ and TranswabTM? Did they make any precaution to ensure that this was not the case?
Regarding culturing:
Two culture media were tested: Dixon agar versus FastFungi medium and the authors find that FastFungi medium is significantly better that Dixon.
- Looking at CFU as a method to investigate which media is the superior is tricky. CFU depend not only on the numbers of colony forming units (conida), but also on how the inoculation was done e.g. a thin layer separate the conida which make it easier to differentia them from each other (to identify each CFU) whereas it is difficult to separate the CFUs in a thick layer. Perhaps the authors would consider this in the discussion.
- The authors do not consider the effect of the (transport)time before the inoculation onto the plates as an explanation for the lower positive rates for the two swabbing methods compared with the gauze there is inoculated immediately?
Author Response
The study is well written and interesting and focus on optimal sampling technique and testing a new culture media:
Response: We thank the reviewer for this positive comment.
Regarding optimal sampling technique:
The authors are testing three methods sterile gauze, dry swab and Transwab and finds that the sterile gauze has a significant higher culture rate than the two others.
- Did the author consider that the area of rubbing with a sterile gauze might be larger than the area swabbed with the ‘dry swab’ and TranswabTM? Did they make any precaution to ensure that this was not the case?
Response: The area of rubbing with sterile gauze might be larger than the areas swabbed with the ‘dry swab’ and TranswabTM. Our study was designed to assess the efficacy of the entire procedure for culturing Malassezia yeasts, but did not aim at controlling for the surface size.
Regarding culturing:
Two culture media were tested: Dixon agar versus FastFungi medium and the authors find that FastFungi medium is significantly better that Dixon.
- Looking at CFU as a method to investigate which media is the superior is tricky. CFU depend not only on the numbers of colony forming units (conida), but also on how the inoculation was done e.g. a thin layer separate the conida which make it easier to differentia them from each other (to identify each CFU) whereas it is difficult to separate the CFUs in a thick layer. Perhaps the authors would consider this in the discussion.
Response: It was impossible to modify the inoculation procedure because the number of CFU cultured was highly variable and unpredictable A maximum of 300 CFU per plate could be counted; in the rare cases were CFU could not be distinguished the CFU number was set to 300. In each plate 1-3 isolated colonies were subjected to the identification procedure (section Methods, lines 98-99).
- The authors do not consider the effect of the (transport) time before the inoculation onto the plates as an explanation for the lower positive rates for the two swabbing methods compared with the gauze there is inoculated immediately?
Response: All samplings were done in the same building were the plates were cultured (IHU MI is a university hospital institute that has both diagnostic and research laboratories). They were thus ino
Reviewer 2 Report
The Authors compared three methods of skin sampling, to detect Malassezia yeasts by culture from the human skin.
The Authors found that rubbing with a sterile gauze was the most efficient method for recovering Malassezia yeast from the skin for culture, with a four times greater positive culture rate by using sterile gauzes than that by using either dry swabs or Transwab.
The authors do not specify what is the extension of the surface sampled with the swabs: a larger surface is sampled using a sterile gauze so it is logical that more fungal cells are taken.
The Author admit that a major limitation of their work is the small number of individuals recruited but they claim that their study design reached sufficient statistical power. I believe that 10 individuals included in the study are really too few to draw reliable conclusions. Other Authors (for example, Pedrosa et al, BJD 2018) have conducted similar studies by recruiting a much higher number of subjects with skin diseases. The Authors should have no difficulty in recruiting healthy subjects.
Author Response
The Authors compared three methods of skin sampling, to detect Malassezia yeasts by culture from the human skin.
The Authors found that rubbing with a sterile gauze was the most efficient method for recovering Malassezia yeast from the skin for culture, with a four times greater positive culture rate by using sterile gauzes than that by using either dry swabs or Transwab.
- The authors do not specify what is the extension of the surface sampled with the swabs: a larger surface is sampled using a sterile gauze so it is logical that more fungal cells are taken.
Response: The surface sampled with the swabs was approximatively the same as the one sampled with the sterile gauze since it was the same sampling site. However, as we have state in our response to the remark of reviewer 1, this study aimed to assess the efficacy of the entire procedure for isolating and culturing Malassezia yeasts. We did not control for the surface size but we tried to sample approximatively identical skin areas.
- The Author admit that a major limitation of their work is the small number of individuals recruited but they claim that their study design reached sufficient statistical power. I believe that 10 individuals included in the study are really too few to draw reliable conclusions.
Response: We acknowledge that the number of individuals is relatively small, but to overcome this limitation, the statistical analyses were based on the number of samples, knowing that 15 samples were collected from each individual. The fact that many tests were statistically significant is at odds with the assertion of an insufficient statistical power.
- Other Authors (for example, Pedrosa et al, BJD 2018) have conducted similar studies by recruiting a much higher number of subjects with skin diseases. The Authors should have no difficulty in recruiting healthy subjects.
Response: Recruitment was voluntary. Some individuals were reluctant or refused to participate in our study because of the relatively high number of sampled that had to be collected. Furthermore, the number of individuals included was balanced by the number of samples collected (15 per individual). Additionally, the workload of culturing 15 samples per subject limits the possibility of including much more subjects given. Finally, we do not think that it is ethical to include a high number of subjects in a study when it is possible address a research question by including a smaller number of subjects.
Reviewer 3 Report
Dr. Abdillaha nd colleagues present here a nice and timely report comparing the efficacy of different sampling methods for Malassezia. Briefly, the authors nicely demonstrate that sterile gauze rubbing smapling and inoculation on FasFund medium is the superior method of all three compared.
I have a few minor suggestion to improve the manuscript:
- Samples were taken from only 10 healthy volunteers, but from different body sites. It is confusing and not clear until one finds hidden in the methods, how finally the analysis ends up with 300 cultures. This should be clarified in the abstract and in the beginning of the results, to avoid misunderstanding.
- The analysis is limited to healthy skin, however this point is already addressed in the discussion. Would it be possible also to elaborate on the results comparing the different body sites?
- Please include in the methods the precise composition of the Schaedler agar, for consistency
- As cultures were grown in parallels and in duplicates, it will be good to include also results on the consistency within the duplicates.
- The discussion would profit from a paragraph addressing the role of Malassezia in inflammatory skin condition.
Author Response
Dr. Abdillah and colleagues present here a nice and timely report comparing the efficacy of different sampling methods for Malassezia. Briefly, the authors nicely demonstrate that sterile gauze rubbing sampling and inoculation on FastFung medium is the superior method of all three compared.
We thank the reviewer for this positive comment.
I have a few minor suggestion to improve the manuscript:
- Samples were taken from only 10 healthy volunteers, but from different body sites. It is confusing and not clear until one finds hidden in the methods, how finally the analysis ends up with 300 cultures. This should be clarified in the abstract and in the beginning of the results, to avoid misunderstanding.
Response: We agree with the reviewer’s remark. We clarified the text accordingly and specified the number of sites (lines 19 and 110).
- The analysis is limited to healthy skin, however this point is already addressed in the discussion. Would it be possible also to elaborate on the results comparing the different body sites?
Response: We agree with the reviewer’s remark that we addressed accordingly in the results section (lines 125 to 129).
- Please include in the methods the precise composition of the Schaedler agar, for consistency
Response: Schaedler agar is a commercial product. We specified the supplier’s references (Sigma-Aldrich, ref. 91019-500G), in lines 76-77.
- As cultures were grown in parallels and in duplicates, it will be good to include also results on the consistency within the duplicates.
Response: There were no duplicate culture; samples were inoculated in parallel onto both media (see section Methods, lines 87-88).
- The discussion would profit from a paragraph addressing the role of Malassezia in inflammatory skin condition.
Response: Thi issue has been addressed following the reviewer’s remark in the Discussion section (lines 180-185)